# Drug Discovery Strategies for Inherited Retinal Degenerations

**DOI:** 10.3390/biology11091338

**Published:** 2022-09-10

**Authors:** Arupratan Das, Yoshikazu Imanishi

**Affiliations:** 1Department of Ophthalmology, Eugene and Marilyn Glick Eye Institute, Indiana University School of Medicine, Indianapolis, IN 46202, USA; 2Department of Medical and Molecular Genetics, Indiana University School of Medicine, Indianapolis, IN 46202, USA; 3Stark Neurosciences Research Institute, Indiana University School of Medicine, Indianapolis, IN 46202, USA; 4Department of Pharmacology and Toxicology, Indiana University School of Medicine, Indianapolis, IN 46202, USA

**Keywords:** photoreceptor cells, retinal ganglion cells, inherited retinal degeneration, retinitis pigmentosa, usher syndrome, optic neuropathy, glaucoma, drug discovery, iPSC, mitochondria

## Abstract

**Simple Summary:**

Inherited retinal degeneration is a group of heterogeneous genetic disorders impairing vision. In these diseases, photoreceptor or retinal ganglion cells become dysfunctional and degenerate. Currently, no cures exist for these devastating blinding conditions. In this review, we will discuss targets and drug discovery strategies for inherited retinal degeneration using animal models, stem cells, and small molecule screening with updates on preclinical developments.

**Abstract:**

Inherited retinal degeneration is a group of blinding disorders afflicting more than 1 in 4000 worldwide. These disorders frequently cause the death of photoreceptor cells or retinal ganglion cells. In a subset of these disorders, photoreceptor cell death is a secondary consequence of retinal pigment epithelial cell dysfunction or degeneration. This manuscript reviews current efforts in identifying targets and developing small molecule-based therapies for these devastating neuronal degenerations, for which no cures exist. Photoreceptors and retinal ganglion cells are metabolically demanding owing to their unique structures and functional properties. Modulations of metabolic pathways, which are disrupted in most inherited retinal degenerations, serve as promising therapeutic strategies. In monogenic disorders, great insights were previously obtained regarding targets associated with the defective pathways, including phototransduction, visual cycle, and mitophagy. In addition to these target-based drug discoveries, we will discuss how phenotypic screening can be harnessed to discover beneficial molecules without prior knowledge of their mechanisms of action. Because of major anatomical and biological differences, it has frequently been challenging to model human inherited retinal degeneration conditions using small animals such as rodents. Recent advances in stem cell-based techniques are opening new avenues to obtain pure populations of human retinal ganglion cells and retinal organoids with photoreceptor cells. We will discuss concurrent ideas of utilizing stem-cell-based disease models for drug discovery and preclinical development.

## 1. Introduction

Retinal neurons receive and process visual cues into neuronal signals and deliver them to the brain [1]. Among those neurons, photoreceptor cells are critical input neurons, converting photo-stimuli into signals that are received, modulated, integrated, and processed by horizontal, bipolar, and amacrine cells. Ultimately, these signals are received by retinal ganglion cells (RGCs), which are terminal output neurons directly connected to the brain. Various visual disorders are characterized by the loss of photoreceptor cells or RGCs. For example, photoreceptor cells and RGCs gradually die in retinitis pigmentosa and optic neuropathies, respectively. In mammals, including humans, those retinal neurons cannot be regenerated naturally [2]. Thus, gradual loss of these cells leads to progressive and irreversible blindness [3,4]. Moreover, functions of those individual neurons are frequently compromised, further impairing vision. This type of functional deficit can occur either concurrently or independently of cellular degeneration. Currently, various efforts are placed on developing therapies for blinding disorders with the aim of restoring the functions or delaying degeneration of photoreceptor and ganglion cells.

At the forefront of human genetics, major advances have been made in understanding the inheritable factors causing degenerative photoreceptor disorders such as retinitis pigmentosa, Stargardt disease, Leber congenital amaurosis, and Usher syndrome (USH) [5,6]. Over 300 genes accounting for more than 1% of the human genome are associated with inherited blindness [5,6]. Most of those genes, when mutated, are causative to photoreceptor degeneration and play critical roles in phototransduction, retinoid transport, and metabolism associated with the visual cycle, photoreceptor structures, and their maintenance [7]. Photoreceptor cells metabolically demand the maintenance of their photoreceptive structure and function. Those energy-demanding processes include outer segment renewal, phototransduction, and generation of dark current [8,9]. Accordingly, disruption of cellular metabolic and homeostasis pathways such as glycolysis and autophagy leads to photoreceptor cell loss [9,10,11,12]. A subset of rapidly progressing diseases, such as Leber congenital amaurosis, is characterized by early dysfunction of photoreceptors and/or their cell death, collectively causing vision impairments at birth [13]. As demonstrated in various animal models, functional defects of photoreceptors are often reversible, and associated cellular degenerations can be slowed by experimental drugs. The biochemical and cell biological basis for those therapies will be discussed in this manuscript.

Optic neuropathies are an important class of retinal diseases causing the death of RGCs, although these neuropathies are less prevalent than photoreceptor degenerative disorders [5]. RGCs receive inputs from intermediate neurons such as amacrine and bipolar cells that process and interpret stimuli from photoreceptor cells. Moreover, a small subset (~1%) of RGCs express light-sensitive protein melanopsin. Those cells, known as intrinsically photosensitive RGCs (ipRGCs), can directly receive photostimuli and convey the signal to the circadian pacemaker of the brain [14,15]. The majority of RGCs transmit signals along their long axons to the lateral geniculate nucleus (LGN), while a minority of them innervate the superior colliculus or other centers of the brain [3]. RGCs are highly energy-dependent and vulnerable to mitochondrial dysfunctions because of their unique morphology and functional requirements [3]. Mutations in the mitochondrial electron transport chain components are strongly associated with Leber hereditary optic neuropathy (LHON) [16]. Mutations in the mitochondrial quality control pathway components are reported for dominant optic atrophy (DOA) [17] and for primary open-angle glaucoma (POAG) patients with normal eye pressure, a condition often referred to as normal-tension glaucoma (NTG) [18]. LHON and DOA are inherited optic neuropathies [19]. Glaucomatous RGC degeneration is often associated with the accumulation of damaged mitochondria, disrupted energy homeostasis, and oxidative stress [20,21,22,23,24,25,26]. In addition to mitochondrial dysfunctions, activation of apoptotic pathways [27] and glutamate excitotoxicity [28,29,30,31] are often observed in glaucoma. Moreover, genome-wide association studies (GWAS) have correlated polymorphisms in 127 loci with a high risk for glaucoma [32]. Many of these genes are unrelated to mitochondrial function and homeostasis and provide additional avenues for pharmacological modulation and treatment of RGCs. Elevated intraocular pressure (IOP) is a major contributing factor to glaucoma, and therefore various therapeutic approaches have been developed to target non-neuronal cells and control IOP, as reviewed previously [33,34]. Here, we will discuss druggable targets for improving RGC survival under pathological conditions. We will also discuss stem cell-based approaches that will provide a deeper understanding of RGC biology and potential identification of targets for the development of RGC protection therapies for the above optic neuropathies.

In this manuscript, we will discuss current drug discovery projects for inherited blinding disorders causing photoreceptor and RGC degeneration. Their pathological mechanisms are relevant to highly prevalent blinding diseases such as age-related macular degeneration, diabetic retinopathy, and glaucoma [33,35,36,37]. For those inherited disorders, cellular, tissue, and animal models recapitulating blinding conditions can be investigated to delineate the molecular mechanisms, pathways, and targets. These models demonstrate predictable patterns of pathologies, which in turn serve as useful pharmacodynamic markers for screening molecules and testing therapeutic efficacies. Ocular tissue is amenable to various interventions, including small molecules, antisense oligonucleotides, adeno-associated viruses (AAVs), and stem cells. For antisense oligonucleotide, gene, and stem cell therapies, we recommend some excellent reviews published recently [38,39,40,41,42,43]. Most gene therapies aim at restoring the gene functions that are lost in patients and are highly specific to the mutated genes they are replacing. This review focuses on the development of small molecule therapies. Instead of addressing mutations in a single specific gene, small molecule therapies are often aimed at modulating general pathways afflicted in multiple genetic disorders and have the potential to treat a broader spectrum of blinding diseases.

## 2. Targets and Drug Discoveries for Modulating Pathways in Photoreceptors and Retinal Pigment Epithelial Cells

*Modulating phototransduction*. Photoreceptor cells function in the first steps of vision, receiving photons and translating their energy into chemical and electrical signals via the phototransduction cascade [44,45]. Defects in the phototransduction components can cause activation of aberrant signaling pathways and photoreceptor cell death. Downstream of rhodopsin-mediated phototransduction, phosphodiesterase 6 (PDE6) is involved in the hydrolysis of cGMP. PDE6 consists of α-, β-, and γ- subunits and mutations of the PDE6 β-subunit gene are found in retina degeneration 1 (rd1) and 10 (rd10) mouse models of retinitis pigmentosa [46]. For preclinical studies, the use of rd1 is limited by its extremely early onset of photoreceptor degeneration, which occurs prior to eye-opening and becomes extensive prior to the time of drug administration. Partly because the rd10 mouse model demonstrates relatively slow degeneration of photoreceptors [46], it is one of the most commonly used inherited retinal degeneration (IRD) models to test investigational drugs. PDE gene mutations are the major causes of retinitis pigmentosa in humans, accounting for ~8% of autosomal recessive retinitis pigmentosa cases [46,47]. Because of diminished activity to hydrolyze cGMP, cGMP concentration is elevated in photoreceptor cells deficient in enzymatic activity, synthesis, or transport of PDE [48]. Defects in PDE synthesis and transport occur by mutations in the aryl hydrocarbon interacting protein-like 1 (*AIPL1*) and *PDE6D* (also called PRBP/δ) genes, protein products of which act as chaperones and transporters of PDE6 holoenzyme, respectively [49,50]. Mutations in other phototransduction genes such as cGMP-gated channel subunits (*CNGA1*, *CNGB1*, *CNGA3*, and *CNGB3*) and ciliary structural genes such as peripherin/rds (*PRPH2*) are associated with elevated cGMP-levels, suggesting this is a common defect among various photoreceptor degenerative disorders affecting photoreceptor functions [51,52].

Elevated cGMP concentration leads to excessive activation of the cGMP-gated channel and cGMP-dependent protein kinase (PKG), leading to photoreceptor degeneration [48,53]. Among them, the cGMP-gated channel is the terminal effector of the phototransduction cascade, and its complete inhibition may cause functional deficits in photoreceptors [54]. Nevertheless, partial blocking of the cGMP-gated channel has been effective in treating the retina [55], consistent with the genetic study in which abolishing cGMP-gated channel expression successfully mitigated photoreceptor cell loss in PDE deficient mice. PKG is an attractive target for therapy because it is not directly involved in the phototransduction cascade but involved in the proapoptotic pathway. To discover small molecules effective in circumventing cGMP toxicity, monomeric and dimeric cGMP analogs were synthesized to target PKG [56]. Using a cell culture system, phenotypic assays were effectively utilized to monitor rod photoreceptor survival and assess the potencies and efficacies of candidate PKG inhibitors [56]. A subsequent study using organotypic culture was useful to test the effect of analogs on the survival of cone and rod photoreceptors [56]. The resulting cGMP-analogs, CN03 and CN238, effectively prevented rod photoreceptor cell death in the *rd10* mouse model [56,57]. These studies also exemplify the effective use of quantitative phenotypic assays for drug development which we further discuss in the section addressing protein destabilizing point mutations (also see Figure 1).

*Addressing Visual Cycle Deficiency.* For photoreceptor cells to maintain their visual functions in ambient light conditions, 11-*cis*-retinal needs to be continuously replenished. Photoactivation of rhodopsin results in consumption of 11-*cis*-retinal, which is regenerated via the visual cycle that takes place in photoreceptor and retinal pigment epithelial (RPE) cells (the visual cycle is reviewed in reference [58]). The most prevalent disease causative gene associated with Stargardt disease is *ABCA4*, which clears retinal from photoreceptor disk membranes and RPE endolysosomal membranes [59,60]. When this gene is mutated, bisretinoids, including A2E, deteriorate vision by accumulating in RPE cells [61]. This is because the formation of bisretinoids is accelerated by the delayed clearance of all-*trans-*retinal, which is itself toxic to photoreceptors [62]. As bisretinoids are byproducts of the visual cycle, decreasing the amounts of retinoids passing through the visual cycle is a viable approach for ameliorating the diseases. RPE65 is a retinoid isomerase responsible for this rate-limiting step of generating 11-*cis*-retinoid and is an attractive target for Stargardt disease. Another attractive target is retinol-binding protein 4 (RBP4), which delivers all-*trans*-retinol (Vitamin A) to the eye via its receptor STRA6 [63,64,65]. RBP4 forms a complex with transthyretin, and this interaction is required to avoid renal clearance of RBP4 [66]. Inhibition of RPE65 or RBP4 would result in decreased flow of retinoids into the visual cycle, leading to less efficient generation of 11-*cis*-retinal. Supporting these therapeutic concepts, the amount of A2E dramatically decreased by genetically downregulating RPE65 or RBP4 [67,68]. Nevertheless, too much inhibition of 11-*cis*-retinal production will abolish the visual function of rod photoreceptors nearly completely, as seen in the *RPE65* knockout mouse. In disease contexts, too much or too little 11-*cis*-retinal is detrimental to vision, and there is significant interest in pharmacologically modulating the visual cycle.

Various strategies have been exploited to ameliorate the toxic effects of retinoid intermediates and byproducts in the visual cycle [69], with a particular focus on Stargardt disease caused by *ABCA4* mutations. *Abca4* knockout mouse serves as a model for testing therapeutics because it demonstrates accumulation of A2E and lipofuscin in RPE cells [70], potentially leading to very slow degeneration of photoreceptors in this model [59]. RPE65 inhibitors, such as retinyl amine and *emixustat*, were designed and studied [71,72]. RBP4 antagonists, including A1120, tinlarebant, STG-001, and fenretinide [73], displace retinol bound to RBP4, inhibit the RBP4–transthyretin interaction [74,75], and promote rapid renal clearance of RBP4 [66]. Another compound developed for Stargardt disease is ALK-001, deuterated retinol which slows the reaction rate of retinoid dimerization via a kinetic isotope effect [76]. In the *Abca4*^–/–^ mouse model [70,75], RBP4 antagonists, RPE65 inhibitors, and ALK-001 block the accumulation of cytotoxic A2E and lipofuscin [76]. These compounds [72] are found in various stages of clinical development for Stargardt disease (NCT04489511 and NCT05244304) [77,78], age-related macular degeneration (NCT03845582), and diabetic retinopathy (NCT02753400) [36,78]. Other than these visual cycle modulators, chemicals that form Schiff base adducts with all-*trans-*retinal directly, instead of binding to target proteins, have been invented to sequester all-*trans*-retinal and counter its toxic effect [79].

Visual cycle modulation, although potentially effective, can compromise vision by decreasing the production of 11-*cis*-retinal chromophores. Therefore, to treat a subset of blinding disorders, other pathways were sought for pharmacological modulation. An unbiased systems pharmacology approach was introduced to identify novel G protein-coupled receptor (GPCR) targets, adrenoceptor α 2C (Adra2c), and serotonin receptor 2a (Htr2a) for treating *Abca4*^–/–^ *Rdh8*^–/–^ double knockout mouse [80] which is a Stargardt disease model demonstrating light-induced rapid photoreceptor degeneration [81]. This light-dependent nature allowed efficient testing of GPCR ligands in promoting photoreceptor survival. G proteins coupled to these receptors regulate adenylate cyclase activity for cyclic AMP (cAMP) production; hence, downstream of these GPCR pathways, adenylate cyclase activity can also be modulated to ameliorate photoreceptor degeneration [81]. Cyclic AMP modulation is potentially effective in addressing other blinding conditions caused by rhodopsin mislocalization, as frequently observed in retinitis pigmentosa patients [82]. Rhodopsin is usually localized to the rod outer segments. In various IRDs, rhodopsin is mislocalized to the rod inner segments, where it stimulates adenylate cyclase through ectopic G protein activation [83,84], leading to cAMP accumulation. Thus, inhibition of the cAMP pathway is a promising route for protecting against retinal degeneration [84]. As mentioned above, for photoreceptor degeneration caused by phototransduction defects, these studies on a Stargardt model recapitulate the pivotal roles of cyclic nucleotides and GPCRs in various photoreceptor degenerative conditions [81]. Structure-based and cell-based approaches have been combined to discover molecules that target and modulate GPCRs [85,86]. We envision such combined approaches can further improve the selectivity of GPCR ligands for treating IRDs.

*Targeting metabolism altered in blinding conditions.* Under normal conditions, photoreceptor cells have a high demand for glucose and depend on aerobic glycolysis for generating energy carriers and macromolecules that are critical for photoreceptor outer segment maintenance [9]. In retinitis pigmentosa, OSs become shorter or lost, compromising the vision of these patients [87]. RPE cells deliver glucose to the photoreceptor cells through their apically localized glucose transporter 1 (GLUT1). Losses of OSs are prohibitory to glucose transport by RPE cells because these cells require OS phagocytosis to maintain GLUT1 at their apical side facing photoreceptor cells [12]. Because of the GLUT1 deficiency, glucose accumulates in RPE cells of retinitis pigmentosa mouse models [12,88]. As glucose is required for photoreceptor metabolism and OS renewal, glucose deficiency further exacerbates OS shortening. By promoting the usage or uptake of glucose by photoreceptors, metabolic reprogramming is an attractive therapeutic strategy for blinding disorders.

Promising targets have been characterized for the metabolic reprogramming of RPE and photoreceptor cells. One such target is the phospho-Akt mediated signaling pathway activated in RPE cells by phosphatidylserine. In healthy retinas, RPE cells phagocytose photoreceptor OS tips presenting phosphatidylserine, activating the pathway [89]. This phagocytic process is compromised in retina degenerative conditions, as mentioned above. Phospho-Akt mediated signaling can be artificially activated in degenerating retinas by subretinal injection of OS fragments which promote the apical expression of GLUT1 in RPE cells [12]. In this pathway, Txnip (α-arrestin) plays a pivotal role because this protein is a negative regulator of glucose transfer and promotes endocytosis of glucose transporters under degenerative conditions [90]. Phospho-Akt phosphorylates and promotes the degradation of Txnip, lifting this negative regulation [90]. Similar reprogramming occurs by activating adenosine monophosphate-activated protein kinase (AMPK), which also phosphorylates Txnip and promotes its degradation [91]. Activation of AMPK is an attractive therapeutic strategy for retinal degeneration, leading to enhanced glycolysis as well as mitochondrial biogenesis [92]. AMPK is activated indirectly by metformin, a diabetes drug that increases AMP:ATP ratio by inhibiting mitochondrial Complex I [93]. Metformin promotes photoreceptor survival in rd10 mice [94,95].

Degenerating photoreceptors can be reprogrammed to utilize glucose more effectively by targeting transcription regulators. Sirtuin 6 (SIRT6) is a histone deacetylase which negatively regulates glycolysis via HIF1α [96]. SIRT6 is a potential target for IRD because glycolysis in photoreceptors is promoted by genetically suppressing this protein, as demonstrated in a recessive IRD mouse model with *Pde6b^H620Q/H620Q^* mutation [96,97]. Several SIRT6 inhibitors have been developed and tested in cancer and diabetes animal models [98], and thus it might be interesting to test them in retina degeneration models, starting from mice with mutations in Pde6 genes such as rd10 and *Pde6b^H620Q/H620Q^*. In summary, repurposing of FDA-approved drugs will accelerate the clinical application of retinal metabolic reprogramming, while newly developed drugs will provide new avenues for direct targeting of metabolic regulatory factors.

*Addressing protein destabilizing point mutations.* Missense mutations are frequently observed for IRDs [5]. They often destabilize the protein products by inducing partial misfolding, ultimately leading to their aggregation or rapid degradation. Protein destabilizing missense mutations are also observed for inherited forms of glaucoma [99] and other genetic disorders such as cystic fibrosis [100]. Moreover, defects in protein folding are common among neurological disorders, including Alzheimer’s, Parkinson’s, Huntington’s, and prion diseases [101], and managing these defects is one of the major goals for treating and ameliorating neurodegenerative disorders.

Protein destabilizing missense mutations in rhodopsin, phosphodiesterase (PDE), RPE65, and CLRN1 genes have been well-characterized [102,103,104,105]. Mutations of the rhodopsin gene are the major causes of autosomal dominant retinitis pigmentosa (adRP) [106]. Among them, P23H mutation is the most prevalent and induces opsin misfolding and destabilization [107,108,109]. Regarding autosomal recessive retinitis pigmentosa (arRP), the *rd10* mouse with a missense mutation in the PDE6 β-subunit gene is one of the best models for recapitulating the disease condition. This mutation does not affect the catalytic activity of PDE6 but destabilizes and decreases the protein amount [105]. Likewise, the most prevalent Leber congenital amaurosis (LCA) causative *RPE65* mutation, R91W, destabilizes RPE65 proteins [110]. The instability of mutant proteins can be caused by compromised interaction with molecular chaperones. *CLRN1* gene encodes a four transmembrane glycoprotein and is associated with Usher syndrome type III (USHIII), which causes impairments of hearing and vision [111,112,113,114]. N48K mutation of the *CLRN1* gene is the most common cause of USHIII in North America and among those of Ashkenazi Jewish descent [111,115]. This mutation introduces defects in its only glycosylation site [103,116]. In general, transmembrane proteins or endoplasmic reticulum (ER) lumenal proteins deficient in glycosylation cannot effectively interact with ER-resident chaperones, calnexin, or calreticulin. Therefore, CLRN1^N48K^ protein is unstable and prone to ER-associated degradation [103]. Ultimately, these proteins are degraded via the ubiquitin-proteasome system and/or lysosomes. As a subset of the above-mentioned mutant proteins is unstable but otherwise functional, their stabilization is a viable therapeutic strategy.

Chemical and pharmacological chaperones have been investigated to address the protein destabilizing effects and ameliorate vision impairments caused by missense mutations. Those chaperones directly bind to misfolded mutant proteins and prevent their premature degradation. Sodium 4-phenylbutyrate (PBA) is a drug utilized to treat urea cycle disorders [117]. PBA also acts as a chemical chaperone and improves the vision of R91W RPE65 knock-in mice [104], which is characterized by compromised 11-*cis*-retinal production and early retinal dysfunction [118]. PBA also demonstrated efficacy in the transgenic mouse model of primary open-angle glaucoma caused by a gain-of-function missense mutation in the myocilin gene (*Tg-MYOC^Y437H^*), leading to misfolding of the protein product in the trabecular meshwork cells and IOP elevation [99]. It has also been shown that a chemical chaperon, trimethylamine N-oxide, restores the stability of misfolded myocilin polymorphism variants to wild-type levels [119]. Tauroursodeoxycholic acid (TUDCA) is a putative chemical chaperone [120] which successfully prevented photoreceptor degeneration in the *rd10* mouse model [121].

While those chemical chaperones nonspecifically bind to various misfolded proteins, pharmacological chaperones targeting specific visual proteins have been developed. Those studies are based on the idea that structural analogs or isomers of 11-*cis*-retinal, opsin’s inverse agonist, would stabilize the protein by binding to the chromophore binding site [102,122,123] One of these ligands, 9-*cis*-retinal, serves as a pharmacological chaperone of opsin mutants [122]. Intriguingly, 9-*cis*-retinal also demonstrates strong potential to compensate for 11-*cis*-retinal deficiency as seen in RPE65-LCA [123,124,125] and is potentially complementary to RPE65-AAV gene therapy. More recently, non-retinoid chaperones of rhodopsin mutants have been investigated and demonstrated success in mitigating photoreceptor loss in P23H rhodopsin gene heterozygote knock-in mouse (Rho^P23H/+^), which is a model of adRP and shows progressive photoreceptor degeneration starting around P20 [109,126]. Thus, both chemical and pharmacological chaperones are showing promise in slowing retinal degeneration in pre-clinical studies.

High-throughput screening (HTS) assays have been employed to seek chaperones or stabilizers of mutant proteins in an unbiased manner. For example, to screen for pharmacological stabilizers of CLRN1^N48K^, cell-based phenotypic assays were developed (Figure 1A,B) [127]. In the first step of this strategy, mammalian transgenic cells stably expressing mRNA encoding CLRN1^N48K^ tagged with an antibody epitope were developed and treated with various small molecules. Relative concentrations of CLRN1^N48K^ protein were quantified by high-throughput immunofluorescence confocal microscopy. This procedure allowed robust identification of small molecules that prevented the degradation of CLRN1^N48K^ protein. In the second step, mammalian cells were engineered to detect and screen out general proteasome inhibitors (Figure 1B). This step is crucial for screening drugs for sensory disorders because general proteasome inhibitors are toxic to sensory cells, including photoreceptor neurons and hair cells [128,129]. Medicinal chemistry efforts resulted in hundreds of molecules analogous to the hit molecule named O03 from the screening [127]. Among these analogs, high-potency molecules were identified by the same primary assay, which provided a dose–response curve for each analog (Figure 1A,C). Leftward shifting of the curve, resulting in lower EC_50_ value and thus higher potency, was recognized as an improvement. In vitro metabolic stability, kinetic solubility, and in vivo pharmacokinetics properties were subsequently characterized for promising molecules.

Those efforts resulted in a novel small molecule, BF844, which successfully mitigated sensory loss in an USHIII animal model which demonstrated progressive hearing loss from P22–70 [127]. In theory, this drug discovery strategy is applicable to any mutated proteins of interest associated with inherited disorders if those proteins are unstable but partially functional. Phenotypic high-throughput assays allow various biological pathways to be interrogated even without prior knowledge of specific target proteins; however, it often yields small molecules with unknown mechanisms of action. Therefore, subsequent target identification is important. Advances in mass-spectrometry-based techniques are paving the way for target identifications [130]. For the target identification of BF844, its biotin-conjugated form was synthesized and utilized for affinity purification of interactive proteins that were subjected to mass spectrometry. Heat shock proteins 60 and 90 were identified to interact with BF844, consistent with the effect of BF844 in proteostasis and preventing ER-associated degradation (ERAD) [127]. ERAD can be modulated in various ways. A recent study indicated that the inhibition of VCP/p97, which is required for misfolded proteins to be extracted from ER for proteasome-mediated degradation, leads to protein stabilization, restoration of outer segment structure, and prevention of photoreceptor degeneration in Rho^P23H/+^ mice and transgenic rats expressing the P23H mutant rhodopsin gene [131]. These findings are encouraging because a pharmacological protein stabilizer does not need to specifically bind to or be tailored for the unstable proteins themselves. By targeting general pathways for protein stabilization, the approaches will have a broad impact in treating various IRDs caused by protein destabilizing mutations.

## 3. Targets and Drug Discoveries for RGC Degeneration

*Mitochondrial homeostasis*. The majority of inherited optic neuropathies are associated with mutations in the mitochondrial homeostasis pathways [19,132]. RGCs are highly ATP-dependent due to the need for varying action potential (AP) firing frequency through the long and partially unmyelinated axons to process visual cues [3]. In most other neurons with long axons, myelination allows lower energy requirements and faster saltatory conduction of APs [133]. In rodents, RGC axons are unmyelinated in the retina through 1mm after the optic nerve head, while human RGC axons remain unmyelinated for an order of magnitude longer distance from the cell bodies until the posterior end of the lamina cribrosa hence require more ATP for AP firing [3]. Electron microscopy analysis revealed high mitochondria density at the human lamina cribrosa compared to the adjacent myelinated portion [134], and a histochemical study showed corresponding high mitochondrial cytochrome-*c* oxidase activity [135]. Therefore, unlike RGCs of rodents and other small animals, high ATP requirements for their unique structure and function make human RGCs highly vulnerable to the dysfunction of mitochondria that are the primary source of ATP. Healthy mitochondria are maintained by the balance among mitochondrial biogenesis, mitophagy (autophagic clearance of damaged mitochondria), mitochondrial fusion, and fission. There is also substantial evidence that proteasomes are involved in the quality control of mitochondrial proteins [136]; however, our recent study indicated proteasomes are not essential for mitochondrial degradation because inhibition of lysosomes but not the proteasomes led to the accumulation of mitochondria and cell death for human stem cell derived RGCs [137].

Mitophagy defects, which result in the accumulation of damaged mitochondria, have been observed in the optic nerve of a murine glaucoma model [25]. In most cases, damaged mitochondria are characterized by depolarized membranes which recruit Parkin/Pink1 for the ubiquitination of outer mitochondrial membrane (OMM) proteins [138,139]. Adaptors are cytosolic proteins that contain a ubiquitin-binding domain (UBD) and an LC3-interacting region (LIR) [138,139]. Upon mitochondrial damage and ubiquitination, adaptors bind to the mitochondria via their UBDs, and the LC3-containing phagophore membrane via its LIR for mitophagy. Optineurin (OPTN), p62 (also called Sequestosome-1 or SQSTM1), Nuclear dot protein 52 kDa (NDP52), Tax1 binding protein 1 (TAX1BP1), and Neighbor of BRCA1 (NBR1) are examples of these adaptors [138,139]. Mitophagy could also occur by a Parkin/Pink1 independent pathway in which receptor proteins that are constitutively localized to OMM are activated upon mitochondrial damage and engage with the phagophore membrane through their LIR for lysosomal targeting [138,139]. BCL2/adenovirus E1B 19 kDa protein-interacting protein 3-like (BNIP3L also known as NIX), FUN14 domain containing 1 (FUNDC1), Gamma-aminobutyric acid receptor-associated protein (GABARAP), GABARAP-like 1, 2 (GABARAPL1, GABARAPL2), and autophagy and Beclin 1 regulator 1 (AMBRA1) are examples of mitophagy receptor proteins [138,139]. Among receptor proteins, NIX-mediated mitophagy is critical for metabolic reprogramming and RGC differentiation during eye development in mice [140]. Neurodegenerative diseases are often associated with mutations in the mitophagy adaptor proteins, such as *OPTN^E50K^* in POAG [141] and *OPTN^E478G^* for familial amyotrophic lateral sclerosis (FALS) [142]. Around 50% of POAG patients were reported to have NTG [143]. E50K frequency was reported to be ~17% among NTG patients [141]. The FALS-associated OPTN^E478G^ disrupts the ubiquitin-binding domain and is found to activate nuclear factor kappa B (NF-κB) mediated cellular inflammation signaling [142].

Tank binding kinase 1 (TBK1) regulates the activity of adaptor proteins, including OPTN [144]. *TBK1* gene duplication mutations have been associated with NTG patients [145]. The lysine residue introduced to OPTN as a result of E50K mutation forms tight ionic interaction with the E698 residue in TBK1 kinase resulting in insoluble aggregates [146,147], potentially inhibiting OPTN function. E50K mutation leads to RGC death, reduction of retinal thickness, and excavation of optic nerve head in transgenic mice [148]. Amlexanox is a potent inhibitor of TBK1 and has been used for treating allergic rhinitis, bronchial asthma, and conjunctivitis in Japan [149]. Oral administration of amlexanox ameliorated obesity and improved insulin sensitivity in mice [150]. In another study, amlexanox was administered orally to *Optn^E50K^* homozygous knock-in mice for 5 months, and their RGC layer thicknesses were assessed by optical coherence tomography (OCT). At 6 months, *Optn^E50K^* homozygous knock-in mice without treatment showed a nearly 15% decrease in RGC layer thickness, while amlexanox-treated mice showed only a 3% decrease compared to the wild-type mice [151]. A small molecule, BX795, was initially developed as a 3-phosphoinositide-dependent protein kinase 1 (PDK1) inhibitor but was also found to be a potent inhibitor of TBK1 in vitro [152]. HEK293 cells expressing OPTN^E50K^ protein formed insoluble aggregates which were dissolved by BX795 [147]. However, the mechanism for BX795 mediated OPTN^E50K^ aggregate dissolution is yet to be discovered. Thus, TBK1 inhibition is an attractive therapeutic strategy for glaucoma treatment.

Defects in mitochondrial fusion are associated with RGC degeneration. Mitochondrial membrane fusion is orchestrated by the fusion of OMM by GTPases Mfn1 and Mfn2 and the fusion of the inner mitochondrial membrane (IMM) by dynamin-related protein Opa1 [153]. Mutations of *MFN1* and *MFN2* genes have been found among POAG patients [18] and mutations of OPA1 among DOA patients [19]. In these diseases, RGC neurons progressively degenerate, leading to vision loss. In addition, a different set of *MFN2* gene mutations was associated with Charcot–Marie–Tooth type 2A (CMT2A) patients with peripheral axonal neuropathy [154]. Thus, loss of Mfn2 function associated with neurodegeneration and restoring Mfn2 function may provide neuroprotection. It was reported that naturally derived small molecule echinacoside increased Mfn2 expression which significantly mitigated cerebral injuries in mice [155]. Thus, pharmacological activation of Mfn2 serves as an attractive therapeutic strategy for neurodegenerative conditions such as glaucoma. Induced pluripotent stem cells (iPSCs) can be derived from patients with mutations in the *MFN2* gene and differentiated into RGCs, which could be an excellent model system for performing small molecule screening and identification of RGC protective compounds, as explained in Figure 2. In addition to Mfn2, nearly 20% of mitochondrial proteins (MPs) have known interactions with small molecules that improve mitochondrial functions [156]. These small molecules could be investigated to identify RGC protective compounds for developing glaucoma therapy.

*Oxidative stress*. Oxidative stress has been identified in the retina of POAG patients with high IOP and experimental glaucoma models [157,158,159]. Peripheral blood cells from patients with inherited optic neuropathies such as LHON and DOA showed increased susceptibility to oxidative stress and apoptotic cell death [160]. Damaged mitochondria are the primary sources of reactive oxygen species (ROS). The nuclear factor erythroid-2 related factor 2 (Nrf2) is the master regulator of ROS clearance. Nrf2 interacts with the Maf transcription factors, and the heterodimeric complex activates nearly 600 genes [161] by binding to the Antioxidant Response Element (ARE) found within their promoter regions [161]. Those genes play roles in antioxidant activities, decreasing heavy metal toxicity, removing damaged proteins, and cellular growth [161,162]. The Keap1-Cullin3 (Cul3)/Rbx E3 ubiquitin ligase systems act as sensors for ROS and regulate the stability of Nrf2. Keap1 forms complex with Nrf2 and serves as a substrate adaptor to ubiquitin ligase Cul3/Rbx, which ubiquitinates Nrf2 for degradation, maintaining a low steady-state level [162]. ROS electrophiles react with the cysteine residues within Keap1 and cause conformational changes in the protein. Thereafter, Keap1 dissociates from Nrf2 and becomes incapable of targeting Nrf2 for degradation. Instead of promoting Nrf2 ubiquitination, Keap1 itself becomes ubiquitinated by Cul3/Rbx for degradation, further decreasing Keap1′s ability to ubiquitinate Nrf2. Stabilized Nrf2 is phosphorylated at Ser40 [163], which is critical for its nuclear translocation and transcriptional activation of the ARE genes. As oxidative stress is associated with multiple diseases, a promising therapeutic strategy is to identify small molecules that will disrupt the Keap1-Nrf2 interaction in the degenerative cells to improve Nrf2-mediated antioxidant and anti-stress activities.

A broad range of Nrf2 activators that target Keap1 and disrupt the Nrf2-Keap1 complex have been tested for neurodegenerative diseases including Huntington’s, Parkinson’s, Alzheimer’s diseases, and multiple sclerosis with promising results [164,165]. High oxidative stress and activation of the Nrf2/ARE signaling pathway were observed in the mouse ocular hypertension glaucoma model [166]. Nrf2 activation by over-expression protected RGCs in mice from the optic nerve crush (ONC) injury [167]. Pharmacological activators that stabilize Nrf2, such as CDDO-Im [168] and dimethyl fumarate [169], have been found to be RGC protective in mice with ONC injury. Nrf2 activators bardoxolone methyl (RTA-402) [170] and sulforaphane [171] are shown to be RGC protective for retinal ischemia-reperfusion injury in rats. As mentioned in the previous section, many of the above neurodegenerative diseases, including POAG, are associated with mitophagy defects with the accumulation of damaged mitochondria. In this pathway, p62 is a critical player for mitophagy. Phosphorylated p62 has been found to interact with Keap1 on the Nrf2 binding site, competing with Nrf2-Keap1 binding [172,173]. Thus, increased free p62 protein level could activate Nrf2 signaling pathways for neuroprotection. The expression level of p62 is also partially regulated by Nrf2 due to the presence of an ARE in its promoter region [174]. To take advantage of this positive feedback regulation, a small molecule screening identified PMI (p62-mediated mitophagy inducer) that disrupts Nrf2-Keap1 interaction and increases p62 expression [175]. PMI enhanced mitophagy of the damaged mitochondria [175], providing dual protection through improved mitophagy and oxidative stress response. Thus, Nrf2 activators such as PMI are strong drug candidates for RGC protection in LHON, DOA, and POAG.

Nrf2 is also degraded by proteasomes via glycogen synthase kinase 3 (GSK3)-mediated phosphorylation and subsequent ubiquitination by the E3 ligase β-TrCP [165]. Thus, GSK3 inhibitors could lead to Nrf2 stabilization and activation to promote oxidative stress response. In fact, several GSK3 inhibitors that activate Nrf2 are under clinical trials for a broad range of diseases [165]. Specific GSK3 inhibitors that confer RGC protection remain unexplored; however, GSK3 is an attractive druggable target for glaucoma therapy.

*Excitotoxicity*. Mitochondrial dysfunction not only compromises ATP production but also elevates the cellular level of ROS, which stimulates excess glutamate release [176]. Glutamate-mediated overstimulation of the N-methyl-D-aspartate receptor (NMDAR) induces excitotoxic cell death in stroke, trauma, epilepsy, Huntington’s disease, and amyotrophic lateral sclerosis [177]. Excess glutamate was observed in the vitreous of glaucoma patients and thought to be the reason for RGC excitotoxic death [178]. However, this observation was later called into question with conflicting evidence [29,30,31]. Nevertheless, it is still believed that minimal elevation of glutamate can cause excitotoxic RGC death [30], as an NMDAR antagonist memantine was effective in protecting RGCs of the high IOP experimental glaucoma monkey model [179]. Likewise, induction of excitotoxicity by NMDA in mouse retinas led to robust RGC degeneration [180]. Hence, NMDAR is a druggable target for RGC protection. Memantine has been approved by the United States FDA for Alzheimer’s disease. A four-year phase-3 randomized double-masked clinical trial with daily oral intake unfortunately failed to prevent glaucomatous progression [181]. Although NMDARs are widely expressed, their number, localization, and subunit composition vary among different types of synapses and neurons [182]. Memantine may be specific to cortical neurons with beneficial effects in Alzheimer’s disease but fail to antagonize NMDAR in human RGCs. For successful glaucoma clinical trials, further studies will be required for the identification of RGC-specific NMDAR antagonists, optimizing dose, duration, and inclusion/exclusion criteria for severity of glaucoma conditions. As we discuss later in this review, human stem cell differentiated RGCs provide an excellent model for drug screening and dose optimization that may lead to successful clinical trials (Figure 2). Stem cell-based approaches could be used for developing RGC-specific therapy to prevent excitotoxicity in glaucoma.

*Cell death signaling*. Glaucoma is initiated at the optic nerve head by the injury to RGC axons, where they exit the eye [3,183]. Downstream of these insults, Jun N-terminal kinases (JNKs) regulate cellular apoptosis [184] and are activated under various neurodegenerative conditions [185,186]. JNK is activated in RGCs following ONC axonal injury [27]. In a separate project, an RNA interference screening was performed on primary mouse RGCs against 623 kinases to identify targets that would promote the survival of RGCs grown in neurotrophin deficient media, which otherwise causes RGC degeneration [187]. This screening identified two kinases, dual leucine zipper kinase (DLK) and mitogen-activated protein kinase 7 (MKK7), which is a substrate of DLK. Knockdown of each of them protected RGCs against the neurotrophin deficient media [187]. DLK activates Jun N-terminal kinases 1–3 (JNK1–3), which are keys for activating cell death signaling [187,188]. As evidence for the proapoptotic role of the DLK-JNK axis, DLK knock-out significantly protected RGCs upon the ONC injury in mice [187]. Thus, DLK inhibition is a viable therapeutic strategy for RGC protection. Small molecule screening with kinase inhibitors identified tozasertib, which inhibits DLK as RGC protective in both in vitro and mouse optic nerve injury models [187]. Tozasertib was originally developed as an aurora kinase inhibitor for cancer treatment [189] but was later found to inhibit multiple kinases, including DLK [187]. ATP-analog kinase inhibitors such as tozasertib target catalytic sites that are structurally similar; hence, they usually target multiple kinases. The safety and efficacy of tozasertib warrants further study before conducting human trials. Nevertheless, DLK is a very promising target for RGC protective drug discovery.

*Use of stem cell-based technologies for drug development*. One of the bottlenecks limiting the development of treatment strategies for RGC degeneration is the challenge of obtaining enough well-characterized human RGCs for research. It is critical to investigate human RGCs for developing RGC protective drug(s) as RGCs are intrinsically very different between mice and humans, both anatomically and genetically [190]. Currently, we have several robust small molecule-mediated differentiation methods for obtaining human stem cell differentiated RGCs (hRGCs) [191,192,193,194,195]. Differentiated hRGCs from patient-derived induced pluripotent stem cells (iPSCs) provide a unique advantage for identifying disease mechanisms and developing drugs relevant not only to the disease-causing mutations but also to patient-specific genetic backgrounds (Figure 2). This approach is key for developing precision medicine personalized for a patient. As RGCs are part of the forebrain, these methods involve forebrain differentiations which yield several cerebral neurons along with RGCs [192,196]. Therefore, a subsequent purification step is necessary for obtaining pure RGCs. To visualize and purify RGCs, a stem cell reporter-based approach was developed where a multicistronic P2A-tdTomato-P2A-Thy1.2 construct was introduced at the end of the *BRN3B* locus using a CRISPR/Cas9-based gene editing technology [191]. BRN3B (POU4F2) is a largely RGC-specific transcription factor which is expressed during the early stage of RGC differentiation and maintained in most of the RGC subtypes [197,198,199]. P2A is a self-cleaving peptide [200] which separates BRN3B, tdTomato and Thy1.2 proteins, allowing them to maintain their functions. The design of this reporter line allows monitoring of successful RGC differentiation by tdTomato fluorescence and immunopurification of RGCs by Thy1.2 protein which is a cell surface transmembrane protein with extracellularly exposed epitope amenable for immunopurification. Using CD90.2 magnetic beads against Thy1.2, obtained hRGCs showed more than 90% purity [137,191]. These cells showed remarkable similarity to human RGCs in their morphological features, genetic profile and electrophysiology [191,192]. Moreover, differentiated hRGCs obtained by the above method can be successfully grafted on mice retinas with neurites projected into the inner plexiform layer (IPL) [201].

With the advent of technology to differentiate stem cells and edit their genes, stem cell-based approaches are becoming increasingly powerful for investigating disease pathology and screening drugs. For example, drug screening on human pluripotent stem cell (hPSC)-derived cortical neural progenitor cells identified drugs for treating Zika virus infection, for which efficacy was further validated in a mouse model [202]. Stem cell-derived organoids were used to test drugs that mitigate the deleterious effects of CTFR gene mutations which cause cystic fibrosis disease [203]. Identification of dose range and treatment duration for drugs in human clinical trials are critical steps, as animal data often do not translate into human outcomes [204]. Stem cell-differentiated models have been helpful in developing HTS paradigms. For instance, hPSC-derived human kidney organoids are used in HTS screening for the identification of effective drugs [205]. Patient-derived iPSCs or CRISPR/Cas9-based gene-edited hPSCs are unique tools for investigating RGC degeneration mechanisms and drug development for DOA, LHON, and POAG with genetic disorders such as *OPTN^E50K^* mutation. Stem cell differentiated RGCs provide an HTS platform to identify compounds that will improve mitochondrial morphology, cellular metabolism, neurite growth, and RGC survival (Figure 2) for developing RGC protective therapies.

## 4. Concluding Remarks

In this review, we discussed ongoing drug discovery projects and cellular processes that provide tractable drug targets for IRDs. We apologize to the investigators who are conducting excellent research in the area of ocular drug discovery, which we are unable to cover due to page limitations. Many of the IRDs share abnormality in the causative cellular pathways. Some of the therapeutic approaches are mutation independent, while others can cover a heterogenous set of mutations in different IRD genes. These efforts will be complementary to gene replacement therapies that focus on addressing mutations of a single gene. Gene and cell replacement therapies are invasive and irreversible. Unlike these hard-to-tune therapeutics, small molecules with well-established pharmacokinetic properties are reversible and tunable and will serve as a safe and noninvasive option for vision disorders.

Inventions of therapies depend on our deep understanding of the affected biological processes in the retina and ocular systems. It has been challenging to reproducibly obtain high-quality retinal neurons from human retinas because they are difficult to culture and often sampled at various post-mortem time points and under different storage conditions. To circumvent the use of postmortem tissues, we now have unprecedented access to pure populations of human RGCs [191] and RPE cells [206] to understand the disease mechanisms and pharmacological targets, thanks to recent advances in methods for stem cell differentiation and subsequent purification. In designing clinical trials, those models will assist pharmacological studies in optimizing dosage regimens and detecting possible adverse effects. To model photoreceptor degeneration, stem cells were successfully differentiated into retinal organoids with photoreceptor cells appearing at the outer surface [207,208]. Stem cells can be genetically engineered so that these differentiated photoreceptors express a green fluorescent protein, which can be utilized to assess photoreceptor survival in response to small molecule treatments [209]. We hope that this stem cell organoid model system will help recapitulate human photoreceptor degenerative disorders that are often difficult to reproduce in rodent models. For example, retinitis pigmentosa seen in Usher syndrome is difficult to recapitulate in rodent models, which in general demonstrate very mild symptoms. In addition to advances in generating retinal organoids, there are encouraging developments in generating new animal models of inherited retinal degenerations. For example, a pig model of Usher syndrome type IC, which demonstrates visual impairments and morphological defects in photoreceptor cells, was developed [210]. Because of the physiological similarities to humans, pig models are more favorable than rodents for preclinical testing of investigational therapies. For glaucomatous RGC degeneration, several new genetically modified animal models have been developed. These models are useful for preclinical studies of small molecules because the onset and duration of their blinding conditions were extensively characterized (for details, please read ref. [211]). Last but not least, an increasing number of therapeutic approaches will modulate retinal physiology as a whole to address degenerative conditions. One example is the metabolic reprogramming of RPE–photoreceptor interactions. Another example is the reprogramming of RGCs by targeting retinoic acid receptors to reduce their noise to compensate for the loss of photosensitivity at the level of photoreceptor cells [212]. Those approaches are robust and likely applicable to a broad spectrum of IRDs.

## Figures and Tables

**Figure 1 biology-11-01338-f001:**
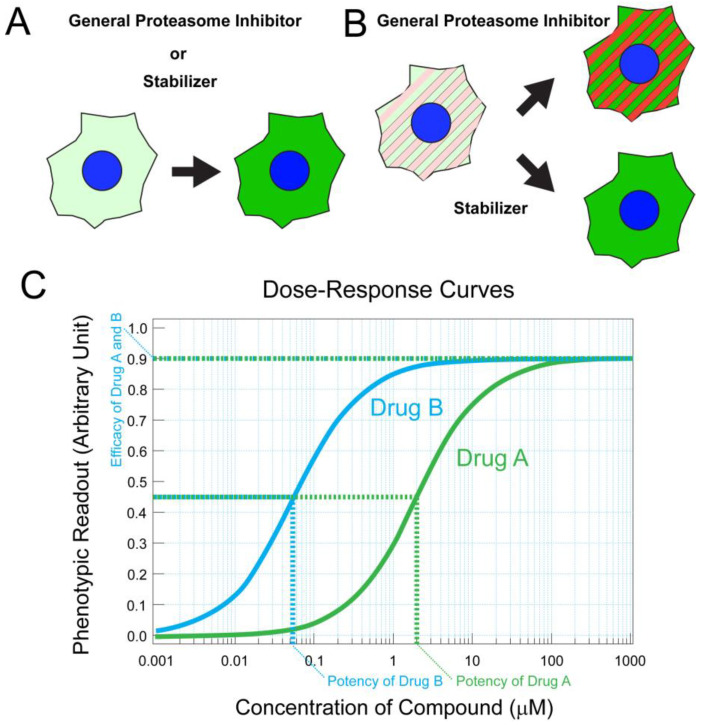
Cell-based assay for drug discovery and medicinal chemistry optimization. (**A**) Drug screenings can be conducted using stable mammalian cell lines that express unstable mutant proteins of interest fused to antibody epitope tags or fluorescent proteins. Using fluorescence intensity (green) as a surrogate, it is possible to measure quantities of mutant proteins. Such an assay can be combined with high-throughput fluorescence microscopy to assess the efficacies of drug-like molecules in stabilizing specific proteins. Protein stabilization will result in increased fluorescence (shown in green). (**B**) An assay was designed to detect and screen out proteasome inhibitors. Red fluorescent protein fused to a strong degron motif was genetically introduced to a mammalian cell line (red). As the degron motif promotes proteasome-mediated degradation, this red fluorescent protein exists in low quantities (left, red). The same cell line was designed to express unstable mutant protein tagged with a fluorescent protein (green) which also exists in low quantities, as explained in (**A**). The cells will demonstrate increased red and green fluorescence when proteasomes are inhibited (right top, red and green stripes). When a drug can stabilize mutant proteins specifically without inhibiting proteasome, the cells will only demonstrate increased green fluorescence (right bottom, green). In principle, similar assays can be developed for any mutated proteins of interest. (**C**) The phenotypic assay can be utilized to derive dose–response relationship curves. For example, average cellular fluorescence intensities obtained from assay (**A**) can be plotted against concentrations of a compound. This dose–response relationship can be utilized for structure–activity relationship (SAR) studies. Improved potency results in a leftward shift of the dose–response curve, while improved efficacy will result in increased maximum effects as measured in phenotypic assay. The strength of this approach is that it can measure variable phenotypic readouts as long as they are quantifiable. For example, the degree of photoreceptor survival can be used as a phenotypic readout for drug screening and optimization.

**Figure 2 biology-11-01338-f002:**
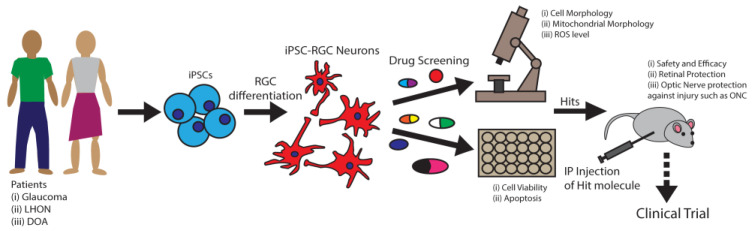
Stem cell differentiated neurons for drug development. RGCs are differentiated from iPSCs which are derived from skin fibroblasts or peripheral blood mononuclear cells of patients. During days 35–45 of differentiation, RGCs (red cells) could be isolated by immunopurification. These RGCs provide a platform for drug screening against disease-associated mutations. HTS, based on microscopic analysis, could identify drugs that improve cellular morphology, mitochondrial morphology, and lower ROS level. Plate-reader-based assays could identify drugs that increase cell viability and lower apoptosis. These compounds are referred to as hits. As a next step, RGC protective hits need to be tested in animal models such as by intraperitoneal (IP) injection in mice for identifying safe dose and treatment duration regimens, then efficacy for protecting RGCs under injury such as optic nerve crush (ONC). This strategy could pave the path for clinical trials in humans with prior data for optimum dose and toxicity in human cells.

## Data Availability

No data have been generated for this article; all the data are cited from published literature.

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
